# Disentangled State Space Models: Unsupervised Learning of Dynamics across Heterogeneous Environments

**Đorđe Miladinović** [*] **& Joachim M. Buhmann**
ETH Zurich, Department for Computer Science
`{djordjem,jbuhmann}@ethz.ch`

**Waleed Gondal** [*] **& Bernhard Schölkopf & Stefan Bauer**
Max-Planck Institute for Intelligent Systems
`{waleed.gondal,bs,bauers}@tuebingen.mpg.de`

## Abstract

Sequential data often originates from diverse *environments*. Across them exist both shared regularities and environment specifics. To learn robust cross-environment descriptions of sequences we introduce *disentangled state space models (DSSM)*. In the latent space of DSSM environment-invariant state dynamics is explicitly *disentangled* from environment-specific information governing that dynamics. We empirically show that such separation enables robust prediction, sequence manipulation and environment characterization. We also propose an unsupervised VAE-based training procedure to learn DSSM as Bayesian filters. In our experiments, we demonstrate state-of-the-art performance in controlled generation and prediction of bouncing ball video sequences across varying gravitational influences.

## 1 Introduction

Learning dynamics and models from sequential data is a central task in various domains of science (Durbin & Koopman, 2012). This includes managing input of diverse complexity e.g. natural language (Graves, 2013), videos (Srivastava et al., 2015) or financial time-series (Øksendal, 2003). It is also crucial for building interactive agents which use reinforcement and control algorithms on top (Finn & Levine, 2017). Traditional choice in engineering are *state space models (SSM)* (Koller et al., 2009), typically found in form of Kalman filters (Gelb, 1974) where well-crafted, relatively simple state representations and (normally linear) functional forms are used. To improve flexibility, new solutions rather learn model-free SSM "from scratch". Due to their non-autoregressive architecture they make an attractive alternative to recurrent neural networks.

Several recent works have already recognized the benefits of introducing additional structure into SSM: the requirement of separating confounders from actions, observations and rewards (Lu et al., 2018) or content from dynamics (Yingzhen & Mandt, 2018; Fraccaro et al., 2017), especially for transfer learning and extrapolation (Kansky et al., 2017). Complementary to these approaches, we focus on learning structured SSM to decouple system *dynamics* into its generic (enviroment-invariant) and environment-specific components. Some examples of sequential data which naturally admit this structure are given in figure 1. Dynamics of these are defined by some constant external factors which we jointly refer to as environment.

More concretely, we explore a panel data setting in which we are given multiple sequences describing the same time-evolving phenomena, one or more per *environment* $e$. We would like to learn a *robust* non-parametric SSM to represent the dynamics of that phenomena across these environments, and robustly extrapolate to the unseen ones. To do so, we explicitly model $e$ as a learnable static element of the latent space. Our idea is based on the assumption that one can decouple sequence

---

[*]Equal contribution

dynamics to: *(i)* the generic part which is *invariant* across environments; and *(ii)* the environment-specific part. In other words, true $e$ integrates all unobserved environment-specific influences which bias generic system dynamics. Our hypothesis is that considering *disentangled*, implicitly *causal* structure of SSM enhances predictive robustness, domain adaptation, and allows for environment characterization and reasoning under interventions e.g. counterfactual inference.

$$E + S \underset{k_1}{\overset{k_{-1}}{\rightleftarrows}} ES \overset{k_2}{\rightarrow} E + P$$ 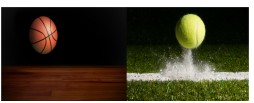 $$\frac{dx}{dt} = \alpha x - \beta xy$$ $$\frac{dy}{dt} = \delta xy - \gamma y$$ 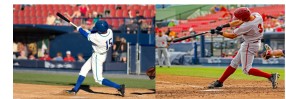

Figure 1: **Sequential systems across environments.** Examples include, from left to right: *(i)* Michaelis-Menten model for enzyme kinetics, governed by reaction rate constants $\vec{k}$; *(ii)* bouncing ball kinematics, determined by ball weight and playground characteristics; *(iii)* ODE dynamics, governed by model parameters; *(iv)* bat swinging motion, influenced by the person performing it. In each example, environments are defined differently, depending on what governs sequence dynamics.

OUR KEY CONTRIBUTIONS

**DSSM.** We introduce a class of non-parametric SSM tailored to exploit invariance from sequential data originating from heterogeneous environments. *Disentangled state space models (DSSM)* (see figure 2d) form a joint environment model while explicitly decoupling what is generic in sequence dynamics from what is environment-specific. This enhances robustness and the ability to extrapolate knowledge to unseen environments.

**Bayesian filtering.** We extend on recent advances in amortized variational inference to design an unsupervised training procedure and implement DSSM in form of Bayesian filters. In the spirit of (Karl et al., 2016), well-established reparameterization trick is applied such that the gradient propagates through time. While VAE heuristic provides no convergence guarantees, it is fast, robust and allows end-to-end training.

**Video prediction and manipulation.** We analyze video sequences of a bouncing ball, influenced by varying gravity (environment). We *outperform state-of-the-art* K-VAE (Fraccaro et al., 2017) in predictions, and also *do interventions* by "swapping environments" i.e. we enforce a specific dynamic behaviour by using an environment from another sequence which exhibits the desired behaviour. Example videos are available at: `https://sites.google.com/view/dssm`.

## 2 RELATED WORK

Closely related to our proposal are approaches which consider structured and disentangled representation of videos, separating the pose from the content (Denton et al., 2017; Tulyakov et al., 2017;

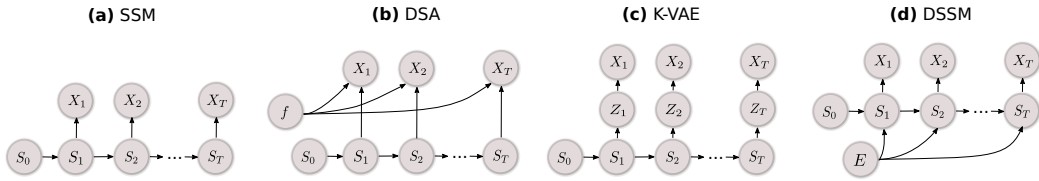

Figure 2: **DSSM and related architectures.** **(a)** Traditional SSM architecture was used e.g. in (Karl et al., 2016). **(b)** Disentangled sequential autoencoder (DSA) (Yingzhen & Mandt, 2018) decouples time-invariant *content* from the time-varying features. **(c)** Kalman-VAE (Fraccaro et al., 2017) separates object (content) representation from its dynamics (we did not depict control input here). **(d)** DSSM introduce environments $E$ to model environment-specific effects on sequence *dynamics*.

Villegas et al., 2017; Yingzhen & Mandt, 2018) or the object appearance from its dynamics (Fraccaro et al., 2017). Proposed models were shown to improve the prediction (Villegas et al., 2017; Denton et al., 2017) and enable controlled generation with "feature swapping" (Tulyakov et al., 2017; Yingzhen & Mandt, 2018). This so called *content-based disentanglement* was also performed in speech analysis where the structure imposed the explicit separation of sequence- and segment-level attributes (Hsu et al., 2017).

VAE frameworks have already been extended to sequence modeling (Marino et al., 2018), and applied to speech (Bayer & Osendorfer, 2014; Chung et al., 2015; Fraccaro et al., 2016; Goyal et al., 2017), videos (Yingzhen & Mandt, 2018) and text (Bowman et al., 2015). However, these (mainly) recurrent neural network-based approaches are autoregressive and hence not always suitable e.g. for planning and control from raw pixel space (Watter et al., 2015; Hafner et al., 2018).

To "image the world" (Ha & Schmidhuber, 2018) from the latent space directly and circumvent the autoregressive feedback, alternative methods learn SSM instead (Karl et al., 2016; Fraccaro et al., 2017; Krishnan et al., 2017). In DVBF (Karl et al., 2016) SSM is trained using VAE-based learning procedure which allows gradient to propagate through time during training. K-VAE by Fraccaro et al. (2017) is a two-layered model which decomposes object's representation from its dynamics. DKF (Krishnan et al., 2015), and very closely related DMM (Krishnan et al., 2017), admit SSM structure but the state inference is conditioned on both past and future observations, so the structure of a filter is not preserved. This is problematic as noted by Karl et al. (2016). Similar issues can be found in (Yingzhen & Mandt, 2018).

As opposed to *content-based* methods which focus on the observation model, our work is focused on *dynamics-based* disentanglement. This makes our approach complementary to existing (see also figure 2). For example, while DSA can represent and manipulate the shape or color of a bouncing ball, our method can manipulate its trajectory. To implement our Bayesian filter, we blended some recent ideas in amortized variational inference (Karl et al., 2016; Marino et al., 2018) and adapted them to fit our novel DSSM architecture.

## 3   VARIATIONAL BAYESIAN FILTERING FOR DSSM

In this work, we assume that the underlying system is deterministic i.e. the latent process noise $\beta$ and observation noise $\omega$ are both uncorrelated in time. We consider the following DSSM description:

$$X_i = g(S_i) + \omega_i, \quad \omega_i \sim N(0, \Sigma_\omega) \tag{1}$$
$$S_{i+1} = f(S_i, E) + \beta_i, \quad \beta_i \sim N(0, \Sigma_\beta) \tag{2}$$

where $f$ and $g$ represent arbitrary flexible functions. $X_i \in \mathbb{R}^D$ represents the observation in time step $i$ and $S_i \in \mathbb{R}^N$ is the corresponding latent state. $\Sigma_\beta$ and $\Sigma_\omega$ are noise covariances which we for simplicity assume are isotropic Gaussian. Our goal is to jointly learn the generative model which consists of the transition function $f$ and observation function $g$, together with the corresponding recognition networks $\phi_\beta^{enc}$, $\phi_E^{enc}$ and $\phi_S^{enc}$ which infer the process noise residual $\beta_i$, the environment $E$, and the initial state $S_0$ respectively. The framework overview is given in figure 3.

**Generative model.**   Given an observed sequence $\vec{X}$ of length $T$, the joint distribution is:

$$p(\vec{X}, \vec{S}, E, \vec{\beta}) = p_0(S_0)p_0(E)\prod_{i=1}^{T} p(X_i|S_i)p(S_i|S_{i-1}, E, \beta_i)p_0(\beta_i) \tag{3}$$

This follows from figure 2d and the assumption that the process noise is serially uncorrelated. We set the prior probabilities of the initial state $p_0(S_0)$, environment $p_0(E)$ and process noise $p_0(\beta_i) = p_0(\beta)$ to be zero-mean unit-variance Gaussian. Conditioned on $\beta_i$ and $E$, state transition is deterministic and the probability $p(S_i|S_{i-1}, E, \beta_i)$ is a Dirac function with the peak defined by equation (2). The emission probability $p(X_i|S_i)$ is defined by equation (1).

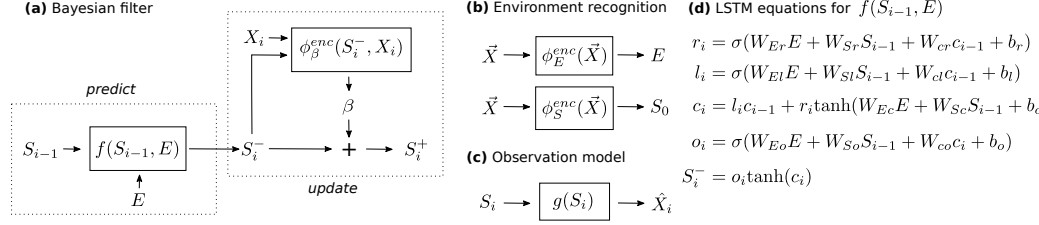

Figure 3: **Variational Bayesian filtering framework.** **(a)** predict-update equations of proposed Bayesian filer: *Predict step* generates the *a priori* state estimate $S_i^-$. *Update step* corrects the estimate ($S_i^+$) using observation $X_i$ through the residual vector $\beta_i$; **(b)** in environment recognition phase the environment $E$ is inferred, and the initial state $S_0$; **(c)** function $g$ maps states into observations; **(d)** an example implementation of the transition function $f$ from equation (2) using LSTM equations (Graves, 2013) (as done in our experiments).

**Inference.** Joint variational distribution over the unobserved random variables $E$, $\vec{S}$ and $\vec{\beta}$, for a sequence of observation $\vec{X}$ of length T factorizes as:

$$q(\vec{S}, E, \vec{\beta}|\vec{X}) = q(E|\vec{X})q(S_0|\vec{X})\prod_{i=1}^{T} q(S_i|\beta_i, S_i^-)q(\beta_i|S_i^-, X_i)q(S_i^-|S_{i-1}, E) \qquad (4)$$

Here, the conditionals $S_i^-|S_{i-1}, E$ and $S_i|\beta_i, S_i^-$ are deterministic and defined by equation (2). The remaining factors are given as follows:

$$q(\beta_i|S_i^-, x_i) = \mathcal{N}(\mu^{\beta_i}, \Sigma^{\beta_i}), \quad [\mu^{\beta_i}, \Sigma^{\beta_i}] = \phi_\beta^{enc}(S_i^-, x_i) \qquad (5)$$

$$q(S_0|\vec{X}) = \mathcal{N}(\mu^s, \Sigma^S), \quad [\mu^S, \Sigma^S] = \phi_S^{enc}(\vec{x}) \qquad (6)$$

$$q(E|\vec{X}) = \mathcal{N}(\mu^E, \Sigma^E), \quad [\mu^E, \Sigma^E] = \phi_E^{enc}(\vec{x}) \qquad (7)$$

**Learning.** To match the posterior distributions of $E$, $S_0$ and $\vec{\beta}$ to the assigned prior probabilities $p_0(E)$, $p_0(S_0)$ and $p_0(\beta)$, we utilize reparametrization trick (Kingma & Welling, 2013; Rezende et al., 2014). This enables end-to-end training. To define the objective function we derive the variational lower bound $\mathcal{L}$, which we consequently attempt to maximize during the training. We start from the well-known equality (Kingma & Welling, 2013):

$$\mathcal{L} = \mathbb{E}_{q(\vec{S}|\vec{X})}[\log p(\vec{X}|\vec{S})] - \mathrm{KL}(q(\vec{S}|\vec{X})||p_0(\vec{S})) \qquad (8)$$

Due to the conditional independence of the observations given the latent states, we can decompose the first term as:

$$\mathbb{E}_{q(\vec{S}|\vec{X})}[\log p(\vec{X}|\vec{S})] = \sum_{i=1}^{T} \mathbb{E}_{q(S_i|\vec{X})}[\log p(X_i|S_i)] \qquad (9)$$

The KL term can be shown to simplify into a sum of the following KL terms:

$$\begin{aligned}\mathrm{KL}(q(\vec{S}|\vec{X})||p_0(\vec{S})) = {} & \mathrm{KL}(q(E|\vec{X})||p_0(E)) \\ & + \mathrm{KL}(q(s_0|\vec{X})||p_0(S_0)) \\ & + \sum_{i=1}^{T} \mathbb{E}_{q(\beta_i, E, S_{i-1}|\vec{X})} \mathrm{KL}(q(\beta_i)||p_0(\beta))\end{aligned} \qquad (10)$$

where we dropped the conditional dependency $\beta_i|S_{i-1}, X_i, E$ in $q$ to ease the notation. Full $\mathcal{L}$ derivation is given in Appendix. Algorithm 1 shows the details of the training procedure for one iteration, for a batch of size 1. The extension to the batch training is trivial.

**Implementation.** We model $[f, g, \phi_S^{enc}, \phi_E^{enc}, \phi_\beta^{enc}]$ as neural networks. In our experiments, $g$ and $\phi_\beta^{enc}$ are convolutional/deconvolutional networks. $\phi_S^{enc}$ and $\phi_E^{enc}$ are given as bi-directional LSTM followed by a multilayer perceptron to convert LSTM-based sequence embedding into $S_0$ and $E$, and $f$ as an LSTM cell as elaborated in figure 3d. LSTM equations are taken from (Graves, 2013).

**Optimization Challenges.** Some performance improvements were observed with an additional heuristic regularization term, which ensures the consistency during the inference of environment $E$. Namely, we penalize the step-wise change in time embeddings produced by bi-directional LSTM used to model $\phi_E^{enc}$, in order to enforce $E$ to remain time-invariant. To that end, we add an additional term to our objective function, the *moment matching regularization* term defined as:

$$MM(\phi_E^{enc}(\vec{x})) = \sum_{i=2}^{T} ||h_i - h_{i-1}||^2 \tag{11}$$

where $h_i$ is the hidden state of the $\phi_E^{enc}$ LSTM cell in step $i$. This idea is related to the approaches based on the maximum mean discrepancy (Gretton et al., 2012). Namely, enforcing equality of consecutive cell states corresponds to matching of their first moments.

Furthermore, similarly to (Bowman et al., 2015; Karl et al., 2016) we used a KL annealing scheme. This was helpful for circumventing local minimum and preventing the KL term to converge to zero too early during the training. The exact details are given in our experiments.

## 4 PREDICTION, MANIPULATION AND GENERATION OF VIDEO SEQUENCES

**Bouncing ball in varying gravity settings.** We test our framework on a 2D bouncing ball problem where the ball kinematics is affected by a varying gravity vector. The idea is to evaluate model robustness across environments – gravitational settings. See also visualizations at: `https://sites.google.com/view/dssm`. Using the physics engine code from (Fraccaro et al., 2017), we simulate video sequences of a bouncing ball. During generation, we randomly change the gravity such that it remains constant within a sequence, but may vary across sequences. The gravity vector takes 4 values, depending on whether the gravity points up, down, left or right. The magnitude is kept fixed. Each video frame is a 32x32 binary image. We generate 16'000 trajectories across 40 time steps for training, and another 2000 trajectories across 70 time steps for testing. We first perform long-term forecasting analysis comparing our method against one of the state-of-the-art approaches, the K-VAE from (Fraccaro et al., 2017). Next, we demonstrate controlled generation, by manipulating video sequences. Effectively we perform interventions by "swapping environments" between video sequences, and similarly we swap initial states. Furthermore, we show the ability to perform uncontrolled generation where both initial state and gravity vector are sampled from a prior. Finally we visualize the environment embeddings to provide further intuiton.

**Forecasting ball trajectory.** We use OpenCV inbuilt functions to detect ground truth ball position $p_t$ in each time frame. The exact algorithm for the position extraction is provided in the Appendix

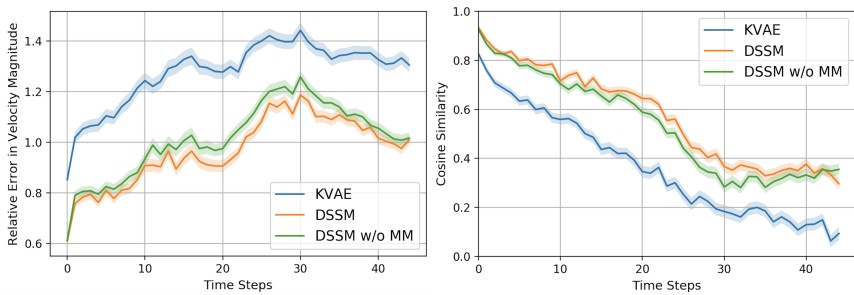

Figure 4: **Bouncing ball trajectory forecasting.** DSSM-based Bayesian filter against K-VAE on the task of long-term forecasting: *(left)* velocity magnitude; *(right)* cosine similarity. *MM* denotes moment-matching regularization. Shown error curves are the test set averages.

B.2. Following (Hsieh et al., 2018; Chang et al., 2016) we define the ball velocity as $v_t = p_{t+1} - p_t$ and compute the relative error in the predicted velocities of the balls for forecasting. Evaluated models observe the first 25 frames of a test sequence and then forecast the next 45. The results for relative error in magnitude and cosine similarity of the velocities are then averaged across all test sequences. This is shown in figure 4. We observe an increase in prediction quality with respect to both metrics in comparison to the benchmarking model K-VAE.

**Video manipulation for controlled generation.** Firstly, the initial state which consists of the velocity vector and the ball position, is extracted from the baseline video sequence and then "injected" into a series of other test sequences. Similarly, we performed the gravity environment replacement. We then enrolled the sequence effectively performing controlled generation (see figure 5).

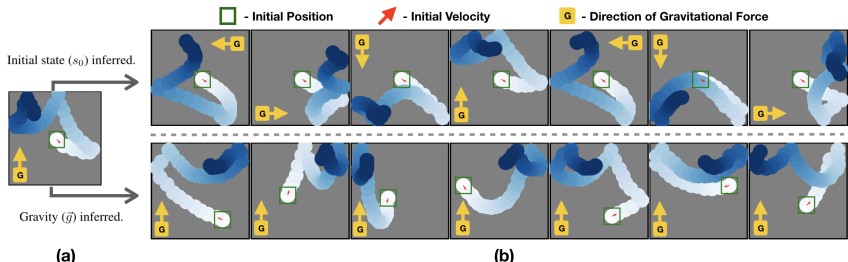

Figure 5: **Environment swapping and controlled generation.** (a) base sequence. (b) test sequences in which we injected (i) environment gravity value; (ii) initial state; of the base sequence inferred using recognition networks $\phi_E^{enc}$ and $\phi_S^{enc}$ respectively.

**Environment identification.** We trained an auxiliary multi layer perceptron classifier to map $E$ to true gravity value. The cross-validation results performed on the training set rendered accuracy of 99.15%. Visualized embeddings (for $E \in \mathbb{R}^3$) are given in figure 6a. Well-defined clusters can be observed, indicating that $E$ indeed represents the true gravity.

**Uncontrolled Generation.** We demonstrate the uncontrolled generation of the sequences where $s_0$ and $E$ are sampled from the priors $p_0(E)$ and $p_0(S_0)$ respectively. In figure 6b we observe how the generated sequences preserve natural bouncing ball dynamics.

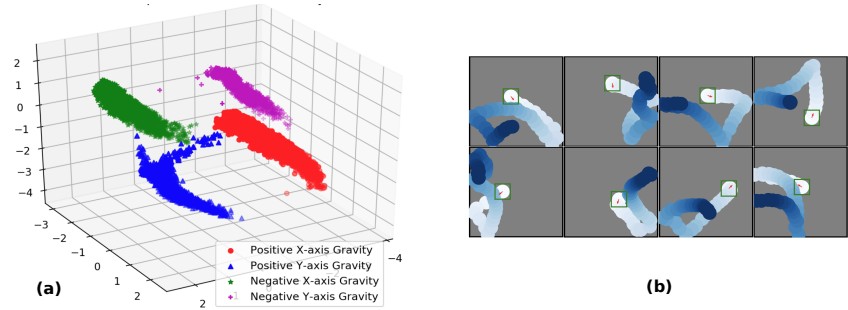

Figure 6: **(a)** learned environment $E$ embeddings. **(b)** uncontrolled video generation.

## 5 CONCLUSION

This work proposes a novel view on data-driven learning of dynamics from diverse environments. We proposed a new class of state space models particularly crafted to exploit this kind of a setting. In disentangled state space models one separates generic system dynamics which is assumed to be invariant across environments and environment-specific information which governs this dynamics. We showed that such separation is beneficial and allows us to learn robust cross-environment models which hold promise to generalize on unseen environments. Our particular application was learning

of the video dynamics of a bouncing ball affected by varying gravitational influences where we achieved state-of-the-art results. Our future work will include other types of data.

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

## A  LOWER BOUND DERIVATION (SECTION 3)

**Conditional log-likelihood term in $\mathcal{L}$**

$$\mathbb{E}_{q(\vec{S}|\vec{X})}[\log p(\vec{X}|\vec{S})] = \mathbb{E}_{q(\vec{S}|\vec{X})}[\log \prod_{i=1}^{T} p(X_i|S_i)] = \mathbb{E}_{q(\vec{S}|\vec{X})}[\sum_{i=1}^{T} \log p(X_i|S_i)]$$

$$= \sum_{i=1}^{T} \mathbb{E}_{q(S_i|\vec{X})}[\log p(X_i|S_i)]$$

*(where the conditional independence follows from the state space model formulation)*

**KL term in $\mathcal{L}$**

$\mathrm{KL}(q(\vec{S}|\vec{X})||p_0(\vec{S}))$

$$= \int_{\vec{S}} q(\vec{S}|\vec{X}) \log \frac{p_0(\vec{S})}{q(\vec{S}|\vec{X})}$$

$$= \int_{E} \int_{\beta} \int_{\vec{s}} q(S_0|\vec{X})q(E|\vec{X})q(\vec{\beta}|\vec{X}, S_0, E)q(\vec{S}_0|S_0, E, \vec{\beta})$$

$$\log \frac{p_0(S_0)p_0(E)p_0(\vec{\beta}|S_0, E)p_0(\vec{S}_0|S_0, E, \vec{\beta})}{q(S_0|\vec{X})q(E|\vec{X})q(\vec{\beta}|\vec{X}, S_0, E)q(\vec{S}_0|S_0, E, \vec{\beta})}$$

*(where we used the factorization of the variational and the prior distribution. $\vec{S}_0$ is vector $\vec{S}$ without $S_0$)*

$$= \int_{E} q(E|\vec{X}) \log \frac{p_0(E)}{q(E|\vec{X})}$$

$$+ \int_{S_0} q(S_0|\vec{X}) \log \frac{p_0(S_0)}{q(S_0|\vec{X})}$$

$$+ \int_{E} \int_{\vec{\beta}} \int_{S_0} q(\vec{\beta}|\vec{X}, E, S_0) \log \frac{p_0(\vec{\beta}|E, S_0)}{q(\vec{\beta}|\vec{X}, E, S_0)}$$

$$+ \int_{E} \int_{\beta} \int_{\vec{S}} q(S_0|\vec{x})q(E|\vec{x})q(\vec{\beta}|\vec{X}, S_0, E)q(\vec{S}_0|S_0, E, \vec{\beta}) \log \frac{p_0(\vec{S}_0|S_0, E, \vec{\beta})}{q(\vec{S}_0|S_0, E, \vec{\beta})}$$

*(where we dropped the integral sums for which the corresponding term does not depend on)*

$$= \mathrm{KL}(q(E|\vec{X})||p_0(E))$$

$$+ \mathrm{KL}(q(S_0|\vec{X})||p_0(S_0))$$

$$+ \mathrm{KL}(q(\vec{\beta}|\vec{X}, E, S_0)||p_0(\vec{\beta}|E, S_0))$$

*(where the last term vanishes since $\vec{s}_0|s_0, E, \vec{\beta}$ is deterministic)*

$$= \mathrm{KL}(q(E|\vec{X})||p_0(E))$$

$$+ \mathrm{KL}(q(S_0|\vec{X})||p_0(S_0))$$

$$+ \sum_{i=1}^{T} \mathbb{E}_{q(\beta_i, E, S_{i-1}|\vec{x})} \mathrm{KL}(q(\beta_i|X_i, E, S_{i-1})||p_0(\beta))$$

*(where we have $p_0(\beta_i|E, s_i) = p_0(\beta)$ by design)*

## B  EXPERIMENTS (SECTION 4)

### B.1  DETAILS

To get compressed representation of each frame, the images are first passed through a shallow convolutional network. Kernel size was set to 3x3, while the network depth was 64. The step size was 1 in both directions. We used ReLU activation units. All of the hidden latent states were equal to 64. To parameterize $g$ we used a deconvolutional network with transposed convolutions. The kernel size was set to 5.

Following the insights from (Bowman et al., 2015), we tried different settings for KL annealing in the model. Since we have three KL terms in our model which have different roles, we do not penalize KL terms of time-invariant components i.e. $KL(q(S_0|\vec{X})||p_0(S))$ and $KL(q(E|\vec{X})||p_0(E))$ as forcefully as $KL(q(\beta_i|S_i, X_i)||p_0(\beta))$ during training. This makes it relatively easier for the model to learn the time-invariant components. Similarly to (Fraccaro et al., 2017), we also found that down-weighing the reconstruction term helps in faster convergence. In particular we applied scaling coefficients of [0.1,0.2,0.3,1.0] for terms $\mathbb{E}_{q(\vec{S}|\vec{X})}[\log p(\vec{X}|\vec{S})]$, $KL(q(E|\vec{X})||p_0(E))$, $KL(q(S_0|\vec{X})||p_0(S))$ and $KL(q(\beta_i|S_i, X_i)||p_0(\beta))]$ respectively.

We use ADAM as the optimizer with 0.0008 as the initial learning rate, and weight decay of 0.6 applied every 20 epochs.

## B.2 ALGORITHM FOR DETECTING BALL POSITIONS

We use OpenCV's (Itseez, 2015) inbuilt functions to detect the pixel level positions of the ball in the images.

```python
import cv2
import imutils
def find_positions(image):
    ret, binary_mask = cv2.threshold(image, 0.01, 1, cv2.THRESH_BINARY)
    binary_mask = cv2.erode(binary_mask, None, iterations=1)
    binary_mask = cv2.dilate(binary_mask, None, iterations=1)
    fake_frame = cv2.convertScaleAbs(binary_mask.copy())
    cnts = cv2.findContours(fake_frame,
                    cv2.RETR_EXTERNAL,
                    cv2.CHAIN_APPROX_SIMPLE)
    cnts = imutils.grab_contours(cnts)
    c = max(cnts, key=cv2.contourArea)
    ((x, y), radius) = cv2.minEnclosingCircle(c)
    return x, y
```

## C TRAINING ALGORITHM (SECTION 3)

---

**Algorithm 1** One iteration of the training procedure

---

**Input:** sequence $\vec{x}$ of length $T$
$[\mu^E, \Sigma^E] = \phi_E^{enc}(\vec{x}), \quad E \sim \mathcal{N}(\mu^E, \Sigma^E)$
$[\mu^S, \Sigma^S] = \phi_S^{enc}(\vec{x}), \quad s_0 \sim \mathcal{N}(\mu^S, \Sigma^S)$
**for** $i = 1$ **to** $T$ **do**
   *Predict step:* $S_i^- = f(S_{i-1}, E)$
   *Estimate residual:* $[\mu_i^\beta, \Sigma_i^\beta] = \phi_\beta^{enc}(S_i^-, X_i), \quad \beta_i \sim \mathcal{N}(\mu_i^\beta, \Sigma_i^\beta)$
   *Update step:* $S_i = S_i^- + \beta_i$
   *Predict observation:* $\hat{X}_i = g(S_i)$
**end for**
ll_loss = LL$(\vec{X}, \vec{\hat{X}})$ (see Eq (9))
kl_loss = KL$(E, S_0, \vec{\beta})$; (see Eq (10))
mm_loss = MM$(\phi_E^{enc}(\vec{X}))$; (see Eq (11))
*Backpropagate(ll_loss, kl_loss, mm_loss)*

---

