# OpenReview forum: "DISENTANGLED STATE SPACE MODELS: UNSUPERVISED LEARNING OF DYNAMICS ACROSS HETEROGENEOUS ENVIRONMENTS"
_ICLR.cc/2019/Workshop/DeepGenStruct — DeepGenStruct 2019_

### Official Review · AnonReviewer2 · 2019-04-06
**Weak evaluation, but reasonable for workshop**

**Rating:** 3
**Confidence:** 2

**Review:**

This paper presents a disentangled generative state space model. By using a global latent variable E the model captures environment-specific information and aims to be disentangled from the rest of the state information of the state space. In a single setting of 2D images of a bouncing ball in a varying gravity settings, this method seems to be yielding reasonable results.

I like the ideas in this paper, and despite a weak evaluation, it is at a reasonable state for a workshop paper.

* Fig 6a suggests that E captures well the direction of the gravity but there seems to be additional information (the variation within Fig6a). Have the authors explored what is the information that creeps in E? And, thus, is the model truly disentagled?

* The video manipulation experiment is great, but seems to be of qualitative nature. It would be nice to see some quantitative results. For example, the same factors could be changed in the fully simulated environment and the output of the video manipulation experiments compared with the ground truth from the simulation.

* Finally, it would have been nice to see more evaluation results on different settings beyond the bouncing ball experiment, both in simulated and real environments. Additionally, a deeper evaluation of the extent to which disentangling happens would be quite useful.



Minor: "This enhances robustness and ability to extrapolate" -> "... the ability"

---

### Decision · Program_Chairs · 2019-04-19
**Acceptance Decision**

**Decision:**

Accept

**Comment:**

Accepted